# Coastal Erosion and Flood Coping Mechanisms in Southern Thailand: A Qualitative Study

**DOI:** 10.3390/ijerph191912326

**Published:** 2022-09-28

**Authors:** Uma Langkulsen, Pannee Cheewinsiriwat, Desire Tarwireyi Rwodzi, Augustine Lambonmung, Wanlee Poompongthai, Chalermpol Chamchan, Suparee Boonmanunt, Kanchana Nakhapakorn, Cherith Moses

**Affiliations:** 1Faculty of Public Health, Thammasat University, Pathum Thani 12120, Thailand; 2Center of Excellence in Geography and Geoinformatics, Department of Geography, Faculty of Arts, Chulalongkorn University, Bangkok 10330, Thailand; 3UNAIDS Regional Support Team for Asia and the Pacific, Bangkok 10200, Thailand; 4Department of Geography, Faculty of Arts, Chulalongkorn University, Bangkok 10330, Thailand; 5Institute for Population and Social Research, Mahidol University, Nakhon Pathom 73170, Thailand; 6Faculty of Medicine, Ramathibodi Hospital, Mahidol University, Bangkok 10400, Thailand; 7Faculty of Environment and Resource Studies, Mahidol University, Nakhon Pathom 73170, Thailand; 8Department of Geography and Geology, Edge Hill University, Ormskirk L39 4QP, UK

**Keywords:** coastal erosion, flood, coping mechanism, community, Thailand

## Abstract

Communities in coastal regions are affected by the impacts of extreme climatic events causing flooding and erosion. Reducing the impacts of flood and erosion in these areas by adopting coping strategies that fortify the resilience of individuals and their localities is desirable. This study used summative content analysis to explore the coping mechanisms of coastal communities before, during, and after various dangers relating to flooding and erosion. The findings from the study show that effective surveillance systems, disaster preparedness, risk mapping, early warning systems, availability of databases and functional command systems, as well as reliable funding are essential to efficiently cope with hazards of coastal flooding and erosion. As flooding and erosion have been predicted to be more severe due to climate change in the coming years, the adoption of effective natural and artificial mechanisms with modern technologies could help coastal regions to be more resilient in coping with the dangers associated with flooding and erosion. Pragmatic policies and programs to this end by actors are critical to averting crises induced by flooding and erosion in coastal areas.

## 1. Introduction

Coastal erosion and flooding occur through three main mechanisms: extreme local sea level rise, powerful wave action, and coastal flooding. This wears down rocks, soils, and/or sands near the coast. Storms and other natural occurrences that result in erosion impact all coastal regions [1]. Since sea levels are rising and socioeconomic growth is increasing the number of people and the value of assets in coastal floodplains, it is anticipated that coastal flood damages will increase dramatically over the twenty-first century. It might inevitably lead to the loss of natural areas, as has already been the case for many years with larger and more devastating environmental as well as human impacts [2]. The perpetual inundation of these zones might gradually result in the disappearance of significant bird and other animal feeding areas and habitats such as salt marshes, mudflats, or beaches and other vital resources. Intertidal habitats are also crucial for commercial fisheries because they serve as vital nursery grounds [3]. Without adaptation, 0.2–4.6% of the world’s population is predicted to migrate each year under 25–123 cm of global mean sea level rise, with corresponding losses to the global GDP of 0.3–9.3%. This makes human reaction critical since coastal erosion will grow as result of sea level rise. To mitigate this effect, it is critical to emphasize the crucial position of long-term coastal adaptation measures [4]. Risk levels of erosion and flooding are increasing in coastal hotspots such as low-lying river deltas due to climate change and local human activities such as river management and urban expansion [5,6,7,8]. Thailand’s geographic position and human activities, like those of other Asian coastal nations, are fast altering the geomorphologic functioning of its delta through a combination of natural and human activity, making it more vulnerable and exposed to dangers from flooding and erosion as sea levels rise globally [9,10]. Although coastal erosion occurs in all parts of Thailand, its rates and potential effects are highly regionalized. Wide beaches and sizable dunes can be severely impacted by erosions and flooding in a single incident [11].

Climate change risks are complex and multidimensional and therefore require multidisciplinary and comprehensive approaches to assessment and adaptation [12,13,14,15]. In the past, more efforts were applied to safeguard the coast by “hardening” the shoreline with levees, seawalls, groins, and riprap. The idea that structural solutions could create more issues than they resolve is becoming more widely accepted as knowledge of how natural shorelines work increases. However, the need for greater resilience of coastal communities, particularly during crises, through the adoption of effective coping mechanisms could be vital in avoiding the catastrophic effects of coastal flooding and erosion [2]. Climate resilience initiatives for coastal communities that adopt efficient coping strategies may be highly cost-effective for localities and globally. Involving multidimensional contributors, such as communities and scientists, in addition to stakeholders will make coping mechanisms for coastal communities in Thailand more resilient [16]. Agreeing on coordinated plans to accomplish community resilience objectives in coping with coastal flooding and erosion by using a participatory and community ownership approach could greatly contribute to a more acceptable and sustainable way of significantly reducing the impacts of climate change and anthropogenic stresses on coastal communities [17]. Adjusting institutional environments to turn impediments into facilitators might provide more immediate approaches to improving adaptable capacity and encouraging efficient use of local adaptation strategies [18]. The combined use of indigenous knowledge and the adaptation of modern technology could boost the capacities of local communities to better deal with disaster-induced challenges in coastal areas [19]. The benefits of adopting efficient strategies to cope with climate change and the associated flooding and erosion by coastal communities may include reducing the effects of storm surges, safeguarding coastal homes and businesses, storing carbon and other pollutants, creating nursery habitats for commercially and recreationally important fish species, and restoring open space and wildlife that support tourism, recreation, and the culture of coastal communities [4]. Hazards associated with erosion and flooding along the coast can have a terrible impact on health. Given their dense population, reliance on transportation, energy infrastructure that is susceptible to flood damage, and high-rise residential housing that may be severely impacted by power and utility outages [20], this could have severe consequences for health and well-being. Mental health effects have been studied in coastal communities that are impacted by the destructive effects of erosion and flooding, and it has been shown that sea level rise can cause emotional and psychological reaction to trauma [21].

Thailand has suffered significant damage from 92 flood events between 1970 and 2022. Affecting 62 million people, the floods caused the death of 4151 people and a direct economic loss of USD 47 billion [22]. This study discusses coping strategies by coastal communities in Thailand that could make them more resilient and can ideally be used regionally, alone or in combination, regardless of country, ethnicity of the coastal community, livelihood activities, type of coast (sandy beaches, mangrove forests, estuaries, and coastal lagoons), meteorology and oceanography conditions, and coastal hazards. This study could help individuals, local governments, and organizations within this common ecosystem to combat climate change, coastal erosion, and flooding in order to improve the health and well-being of coastal communities.

## 2. Materials and Methods

### 2.1. Study Area

Qualitative data were collected from a variety of respondents across two coastal provinces of Thailand. Krabi and Nakhon Si Thammarat are 2 of the 24 coastal provinces of Thailand. As shown in Figure 1, the two provinces located in the southern region of Thailand cover coastal zones of 508,540 acres. Krabi lies on the Andaman Sea coast, while Nakhon Si Thammarat lies on the opposite side, the coast of the Gulf of Thailand. It covers a total coastal zone of 683,392 acres [23]. Shoreline change along the Krabi coast varies between −34.5 to +21.7 m/year in the mangroves and −4.1 to +4 m/year on sandy beaches, while shoreline change along the Nakhon Si Thammarat coast varies between −66 to +16.4 m/year in the mangroves and −22.2 to +10.6 m/year on sandy beaches [24]. As of December 2021, the population of Krabi and Nakhon Si Thammarat is 479,351 and 1,549,344, respectively [25]. The study area consists of two subdistricts in Krabi (Mueang Krabi and Ko Lanta) and two subdistricts in Nakhon Si Thammarat (Pak Phanang and Hua Sai), which exhibit a coastal erosion rate greater than 1 m/year, as shown in Figure 2.

### 2.2. Data Collection

Focus group discussions (FGDs) were conducted with five groups of participants in each province, with each group consisting of six members. Table 1 displays the total number of participants who provided information through focus group discussion for this study. A semi-structured guide was created that consists of questions about (1) the impacts from coastal hazards, (2) control measures, (3) mitigation measures, and (4) emergency preparedness and response. For emergency preparedness and response, the questions covered eight areas: (1) disaster preparedness, (2) standard operating procedures, (3) emergency financial, (4) response and preparedness capacity, (5) information management and communication, (6) risks and assess vulnerability, (7) incident command system, and (8) early warning and surveillance system.

### 2.3. Data Analysis

Summative content analysis is a widely used qualitative research technique within health and social scientific contexts [26]. This approach can explore the concerned aspect and deliver simple insights into the usage of certain words for better understanding and interpretation of the underlying context [27]. A summative approach was used to analyze the findings from the FGDs that consist of four items: impacts from coastal hazards, control measures, mitigation measures, and emergency preparedness and response.

## 3. Results

Coastal erosion and flooding are hazards and generate significant impacts on health and well-being, agriculture, and fisheries in Krabi and Nakhon Si Thammarat provinces. Local coping mechanisms and challenges due to climate change are shown in Figure 3.

### 3.1. Impacts from Coastal Hazards

Coastal erosion and flooding have become a perennial problem due to anthropogenic pressures and climate change, which have made coastal risks such as sporadic coastal erosion and flooding worse and more frequent. In addition, a lot of construction work is going on in the area, which loosens the soil, leaving it vulnerable to erosion in the event of floods. Due to eco-tourism, frequent boat trips to different islands result in water waves now and then, resulting in erosion. In Krabi province, approximately 100 boats operate daily. The result is not only coastal erosion but also the destruction of fish and aquatic life because of fuel from boats. Fishery boats are smaller and use less oil compared to tourist boats.

In the past, there were thick and dense forests around the coastal areas. Because of severe coastal erosion from the sea, the forests including coconut trees were destroyed. The sea is continuously expanding, and local people can no longer enjoy the benefits of windbreak and shade as well as the economic value from the coconut trees as they used to do before.

### 3.2. Control Measures

Illegal fishing activities using illegal tools and equipment have depleted the numbers of fish and destroyed the coral reef. The use of dragging tools is destroying the habitat of shrimp and fish. Fishery officers at the provincial level in Krabi province are implementing an artificial coral reef project. The project sought to provide additional substrate to allow corals to form, creating a new habitat for marine life. Local fishery networks and civil society organizations previously supported the artificial coral reef project. In the past, the artificial coral reef used concrete with 5% coal ash, but recently, a different material with a coal ash composition of around 35% was used. Of major concern to the community was the fact that the coal ash had mercury, and therefore contaminated the fish in the sea. CSOs advocated for the need to simultaneously protect the resilience of the sea while protecting against coastal erosion. Local fishery networks and CSOs protested against the artificial coral reef project and were at one point involved in a legal battle with government officials.

In2013, a conservation group was set up in Nakhon Si Thammarat to conduct patrols and monitor illegal fishing activities during the night; otherwise, specific species of fish would become extinct. In Nakhon Si Thammarat province, the conversation group comprises about 60 active members from about 1000 residents. During storms, it became very difficult to go to the sea to check for any illegal activities. The conversation group made use of bamboo to protect the habitat of the fish, and as a barrier to prevent bigger boats from fishing in specific areas. The challenge, however, was that bigger boats often had cameras and drove fast, which made it difficult for officers to apprehend illegal fishermen. It was highly likely that some locals connived with illegal fishermen and shared with them information regarding patrols and surveillance by fishery officers. At one point, fishermen operating smaller boats gathered weapons in preparation to fight off illegal fishermen, but the fisheries officers did not like the idea of different groups resolving issues through violence.

In the past, people used a soil line to protect themselves from coastal erosion, but this did not solve the problem effectively. The Mangrove Resources Conservation Center advocated for the use of natural means to mitigate coastal erosion, e.g., forest plantations, bamboo, and coconut leaves for collecting clay. Mangrove forests in both Krabi and Nakhon Si Thammarat provinces are conserved by the community themselves. In Krabi province, the RAK Thai Foundation assisted people to plant big trees, e.g., *Ceriops decandra*, to slow down the water waves. This method had the full support of the local administration and produced good results. Some areas made use of the *Pandanus tectorius* trees, whose roots greatly slow down sea waves. Other areas made use of sea grass to slow down waves, and in 2018, this project received support from different organizations. The project was implemented in areas that did not have mangrove forests. The participants argued that sea grass helps in slowing down waves from boats, but not from storms. As such, the priority would be to increase the area covered by mangrove forests, or the construction of waterways to protect villages from the impact of waves. In 2020, a survey to assess the resilience of the area covered by sea grass was planned in Krabi province.

Bamboo was used to slow down sea waves and protect fish habitats. However, the participants felt that bamboo does not offer much protection in curbing coastal erosion. Using bamboo seemed to increase the amount of sand on the shoreline. Use of bamboo was considered beneficial but resulted in some injuries as people tried to access the sea. In 2019, the conservation groups received a budget of around 200,000 THB (US$5500) from the government for the first time to buy bamboo equipment to protect the fish habitats. The budget was given to three provinces: Nakhon Si Thammarat, Songkhla, and Krabi. More money would be needed to expand the area under protection from illegal fishermen. The conservation groups planned to construct mini surveillance houses, but the available budget was just not enough. At times, the officers had no other option but to sleep in small boats when conducting night-time surveillance.

Hard construction remained the government’s preferred control measure against coastal erosion. Communities were against such plans, highlighting that if erosion shifted to unprotected areas, then the project would be way too expensive. Any hard construction around the sea needed permission from the Marine Department. At times, the department did not have officers to survey the area and provide approvals for hard construction projects. Municipality provided sand, and communities helped in packing sandbags. During the monsoon season, people used sandbags to protect the roots of trees and prevent them from falling over. During the huge tropical storm of precipitation anomaly that occurred in 2011, big trees from the mountains were uprooted and swept into the sea. This made it very difficult to catch fish, thereby affecting people’s livelihoods. The positive impact, however, was that the trees formed a natural habitat for fish in the sea, allowing them to multiply exponentially. Communities at the ports used sandbags to protect themselves and their infrastructure against waves. To them, rock construction and digging a canal would offer better protection against waves generated by boats.

In Krabi province, block ways and construction of concrete blocks at 45 degrees were some of the measures used to curb coastal erosion. The block ways protected the beach from waves, thereby allowing tourists to feel safer and visit the area throughout the year. The participants felt that the government needed to give full power to local administrations so that they developed their own areas. The public needed information on any development projects taking place in their area. In most cases, the government only provided information to provincial and district level, but then the information was not cascaded down to communities and villages.

### 3.3. Mitigation Measures

In past flooding situations, the government provided cars to ferry women and children to secure locations, such as the gymnasiums in schools. The participants, however, bemoaned the slow response by the government when responding to disasters. When everyone sought shelter at the gymnasiums, the food was enough in most cases. Affected persons received canned foods. Many organizations provided food hampers, e.g., Red Cross. After the 2019 Tropical Storm Pabuk, which had a wind speed of around 80 km/h, villagers spent almost 2 days without electricity and drinking water before the government chipped in to help. Once the winds were no longer strong, people took boats back to their homes. Non-governmental organizations worked faster than the government and helped in constructing houses. The local administration also assisted in the reconstruction of some damaged houses.

Following the floods in Nakhon Si Thammarat province after the Storm Pabuk in January 2019, government and private sector entities visited the area to assess the situation. Some participants indicated that not every household benefited from the government’s compensation. They stated that about 70% of the affected people received different amounts in compensation for their loss. Some received 6000 THB (US$165), others 11,000 THB (US$302), while some received nothing at all. Some people who owned houses in the affected area benefited from government assistance even though they lived in Bangkok. The participants urged the government to pay at least 5000 THB (US$137) to all affected households since the level of suffering was considered the same for all. In Krabi province, the government and foundations provided compensation to people whose boats were damaged. The participants felt that the process was not transparent, as some poor people whose boats were damaged did not receive anything. This resulted in some conflicts due to a lack of trust between different groups of people following receipt of goods from agencies.

In 2011, agriculture was affected badly by the floods, especially in areas along canals. Some people cut trees in the mountains, but during the flood, their houses were destroyed and the logs were carried away with the water. The first responders to the disaster were from the private sector. After the disaster, several concerts were held to raise donations to assist those that were affected. While the government compensated farmers, representatives from CSOs in Nakhon Si Thammarat bemoaned the lack of any form of compensation to the fisheries sector. The CSOs attributed this discrepancy to the fact that farmers had licenses to their land, while fishermen only had licenses to their boats, and not the sea. In addition, records of farmers existed in the government database, yet there was no database for fishermen. In the beginning of 2019, local fishery networks and CSOs in Nakhon Si Thammarat commenced registrations for all fishermen in the area. According to representatives from the central, regional, and local government, there was a budget for compensation earmarked for the Fishery Office. First, the Fishery Office needed to make a formal report. Second, the disaster impact was supposed to be surveyed to get all the details. Third, regulations under the Ministry of Finance were to be applied in providing compensation to fishermen.

The environmental health group is also actively involved in disaster management in areas such as promotion of environmental health and conducting health impact assessments. During the disaster, they assisted in referring patients by EMS. After the disaster, they engaged in solid waste management and water management following some established guidelines. The environmental health group worked with village health volunteers to educate the residents and communities on what to do before, during, and after the disaster. Under the leadership of the provincial governor, the Public Works and Town & Country Planning Office has also been a major stakeholder in disaster response and management. In previous disasters, they assisted with equipment, e.g., toilet cars, boats, drinking water cars, fire extinguishers, among others.

### 3.4. Emergency Preparedness and Response

#### 3.4.1. Disaster Preparedness

The local administration was mandated to conduct trainings on emergency preparedness, and under normal circumstances, such trainings were conducted once annually. The trainings targeted leaders (village heads and members of local administration sub district) and were cascaded down to the public.

Once informed about an imminent flood, people in the affected area prepared themselves to be evacuated to stay on higher ground. Knowing that a storm was imminent, people grab a few possessions, including blankets, before going to an emergency shelter. Local government officers collaborated with village leaders when moving people to an emergency shelter to ensure that those in most need got the necessary support. The participants bemoaned the lack of money and resources to construct stronger houses. Most people in the area worked as fishermen and made use of local knowledge and experience to escape from storms.

#### 3.4.2. Standard Operating Procedures (SOPs)

At the central or regional level, there are established guidelines on managing a disaster situation, not SOPs. Civil society organizations as well as academia were involved in triangulating data and setting up necessary guidelines to address specific issues. Emergency operation centers (EOCs) were set for each province. The Ministry of Interior then assumed a leadership role, overseeing how all other organizations operate during disaster response. This set up allowed for information sharing among organizations.

#### 3.4.3. Emergency Financial

About 20 million THB (USD 0.55 million) is allocated for disaster management annually, and this is a central budget. Activities under this budget are in sync with those of the Public Works and Town & Country Planning Office, as well as the local administration. Challenges to do with late disbursement of funds to local administration and municipalities greatly affected initiatives to construct walls to protect land from coastal erosion in Krabi province. Representatives from NGOs and media organizations argued that the central budget was too small to have any meaningful impact on the ground. The projects took many years to complete due to limited budget. During and after a disaster, soldiers were the most active people who assisted in food delivery logistics and repairing damaged houses. Representatives from the communities in Nakhon Si Thammarat felt that it is more prudent to allocate the disaster response budget to soldiers rather than to the Department of Disaster Prevention and Mitigation.

The Ministry of Public Health has a budget for disaster management covering issues including chemical and water management, among others. Some agencies, e.g., the Ministry of Natural Resources and Environment, used their own budget availed from time to time by the director of the office. Communities mobilized funds for bamboo projects, mainly from volunteers, but also received some money and other forms of support from government agencies and the private sector. In 2019, communities in Nakhon Si Thammarat for the first time received a budget of 200,000 THB (USD 5500) earmarked for the bamboo project. Residents hope that if more money could be availed, then the area under protection could be expanded.

#### 3.4.4. Response and Preparedness Capacity

At central, regional, and local government levels, the Public Works and Town & Country Planning Office conducts disaster preparedness trainings annually. The Ministry of Public Health also conducts preparedness trainings at provincial levels, and these trainings benefit many other organizations that are invited. The local administration conducts some disaster preparedness trainings as and when funds are available. Other organizations, however, do not conduct trainings due to limited budgets. Instead, they just conduct meetings when disaster strikes. In Krabi province, the Office of Disaster Prevention and Mitigation facilitates trainings, leveraging on the existing curriculum and volunteers. Recently, the office concluded some trainings while leading a project on landslides with funding from the Japan International Cooperation Agency (JICA).

At the community level, members in Krabi province highlighted that most people in the area understand their roles when a hydrometeorological disaster strikes. Trainings at the community level are rarely conducted, mainly because there are always a lot of tourists in the coastal areas. NGOs and media organizations also confirmed that no trainings for tsunami have been conducted, at least in the last 3 years.

#### 3.4.5. Information Management and Communication

At central and regional levels, the Public Works and Town & Country Planning Office in Krabi province has a rich database with regards to which organization is responsible for which activity when it comes to disaster preparedness and response. GIS maps were used in analyzing coverage of activities by organizations, as well as identifying vulnerable groups and locating available emergency shelters. It was reported that in Nakhon Si Thammarat province, updated information on the numbers of patients was stored in a database. The Public Works and Town & Country Planning Office in Krabi province also has a 5- to 10-year roadmap with details on their plans regarding disaster preparedness and response. The main challenge was that while all organizations had their own mission and vision, they were often not aligned and did not speak to each other.

Other local organizations, e.g., RAKS Thai Foundation, helped in collecting data on various issues relating to individuals, households, and their properties and businesses. Over the years, they managed to set up a database, making it easy to conduct some analysis, draft disaster prevention plans, and create awareness among the local communities. Most of the data collected by NGOs and media organizations was relayed to the central levels. At the community level, community leaders kept a list of people in their jurisdiction. Hotel owners also maintained records of their guests.

Residents in the coastal provinces of Krabi and Nakhon Si Thammarat mostly received news and information on disasters and storms through radio and television broadcasts. Various groups, including government agencies, used direct messaging and social media platforms such as Facebook and LINE to disseminate information. However, not everyone had a mobile device compatible with some of the applications. If one fisherman knew about a disaster announcement, he or she disseminated the information to others. In the event of an imminent disaster, it was the responsibility of the Marine Office to inform every boat and every drilling rig on the sea. They anticipated the height of waves and prepared to evacuate people.

In the face of an imminent disaster, announcements to evacuate people came from the provincial/district level down to the local government level. The local government then announced what the people were expected to do. If there was any road damaged by coastal erosion, the communities were urged to inform their local administration authorities so that the area was fixed promptly to facilitate smooth operations during evacuation processes.

#### 3.4.6. Risks and Assess Vulnerability

The Office of Disaster Prevention and Mitigation has a risk map showing details regarding the high-risk areas. Because of repeated disasters in the coastal areas, the exact locations with vulnerable populations are known. With the help of community leaders and village health volunteers, health centers in the coastal areas also have information about patients and risk groups. While information on the numbers of vulnerable people was available, representatives from CSOs in Nakhon Si Thammarat province emphasized the need to have such information integrated into disaster management.

#### 3.4.7. Incident Command System (ICS)

A clear command system is functional in both coastal provinces of Krabi and Nakhon Si Thammarat. At the central level, the Public Works and Town & Country Planning Office has a committee and a command system, which clearly articulates who does what during disaster response. Commands are passed in a top-down approach, and information flows from the provincial level to the local administration level. If an area falls outside the jurisdiction of a local administration, then the municipality would be responsible. It is then the duty of the local administration/municipality to make formal announcements regarding the evacuation procedures to the community in the event of an imminent disaster. Local television and radio stations are used to disseminate the information widely. Where available, tower broadcasts are also used in making announcements to the community so that they prepare for evacuation to emergency shelters and have their identity cards in place. NGOs and media organizations flagged that commands from government agencies take time to get to the community, and response is often slow.

#### 3.4.8. Early Warning and Surveillance System

It was reported that the government had in place an early warning system for tsunamis. This system, which is a loud sound, was tested annually. However, in previous years, the early warning system was not tested during the tourist season for fear of scaring them away. Warning posters were also strategically positioned to show directions for ease of navigation during evacuation. The posters showed the actual locations of temples and schools to use as emergency shelter. Tourists seemed more knowledgeable about the warning posters compared to their Thai counterparts.

In Krabi province, NGOs and media organizations professed ignorance on the availability of any disaster surveillance system, indicating that they only received information and released it to the public. In Nakhon Si Thammarat province, a drug control center ran a surveillance network that allowed them to know which area needed assistance in the event of a disaster. Their surveillance network was quite strong and used mobile equipment, including saw machines and scuba diving suits, to assist children during flooding. The surveillance network leveraged on many volunteers who took part in the training of communities regarding disasters and emergency evacuation.

At central and regional levels, preparedness surveillance is conducted regularly, and after a disaster, reports on the loss and disaster impact are shared with the district offices. The Public Works and Town & Country Planning Office also emphasized that their role entailed monitoring after a disaster. For example, after the Storm Pabuk in 2019, which was made by the mud slide due to the illegal forest activities in these coastal regions, the Public Works and Town & Country Planning Office assessed the infrastructural damage and loss. The Marine Office at the provincial level does not conduct surveillance for coastal erosion. Most of their activities entail surveillance after coastal erosion has been reported so that they design solutions for protection in the future. With regard to flooding, the Marine Office follows reports from the Hydrographic Department and other organizations, and then requests for assistance depending on the nature of the disaster.

It was also reported that health centers under the Ministry of Public Health used Emergency Medical Services (EMS) teams linked from one area to another. Together with community leaders, they conducted area-based surveillance observing the water levels near the mountains.

## 4. Discussion

Based on the long-term recollections and experiences of the communities, this study aimed at identifying concepts and determinants of well-being at community, family, and individual levels. A qualitative study approach is used to provide a thorough knowledge of the vulnerability (exposure, sensitivity, effects, and responses) of coastal communities and help provide a framework for dialogue between officials, NGOs, and community members. Our findings indicate that coastal communities in these areas have a thorough understanding of the state of their coastal environments. They recognize the coastal environments as assets and acknowledge the dynamics of coastal changes as they relate to both natural and societal influences and how the effects of flooding and erosion impact them variously. Communities realized the need for more medium- to long-term investments in the availability of databases, surveillance systems, information sources for disaster communication, early warning, risk mapping, disaster preparedness, training on emergency preparedness, budget for disaster response, command systems, and assistance during and after floods to lessen their vulnerability to flooding and erosion. This is consistent with a previous study suggested that coastal hazards and the nature of the phenomenon are linked to the choice of coping strategy that is made in reaction to the possibility of a natural disaster; therefore, timely and effective sharing of information specifically tailored must be adapted [28]. Moreover, as uncovered through this study, efforts by actors to enhance the ability of the communities to take more preventive and constructive action are based on a clear capacity for experiential learning. Managing the complexity of several interacting environmental and climatic challenges will, however, necessitate extensive cooperation between official and informal organizations at various levels and the creation of common coherent adaptation strategies [29]. No matter how crucial the role of local officialdom, civil society organizations, and communities, these players hardly address the underlying reasons of vulnerability to improve coping mechanisms [30]. To better understand a community’s ability to adapt to, respond to, and recover from hydrometeorological risks, accurate measurement of that community’s circumstances prior to a negative event is necessary. Key considerations include applying lessons learned from both historical experience and contemporary awareness as the adoption of effective coping strategies to flooding requires greater integration of existing knowledge [31]. There is a need for more investments in disaster preparedness, standard operating procedures, emergency financial response and preparedness capacity, information management and communication, risks and asset vulnerability, incident command systems, and early warning and surveillance systems, but these communities typically manage climatic variability in a short-term, reactive, and mostly poorly planned and coordinated manner, with certain efforts such as sea wall construction, which have been reported to be ineffective, as was the case for the 2011 Great East Tsunami in Japan [32]. Some coastal communities in Puerto Rico placed a premium on various aspects of social capital as community-based organizations offer a foundation for members of a group to gather with the shared goal of evaluating their main challenges in terms of hazards exposure and using their social capital and social organization to look for alternatives and reduce vulnerabilities in terms of disaster response and recovery [33]. However, the participants were able to acknowledge their accomplishments and shortcomings, developing an experiential and reflective knowledge system that can lead to learning feedback loops that reinforce and improve adaptive ability. Some actions taken over time are obviously ineffective, inappropriate, and unsustainable. While some are somewhat effective, at least with regard to people and households, others are not, indicating deficiencies that adaptation needs to solve for more resilient coping mechanisms to erosions and flooding in coastal regions.

Faced with frequent hydrometeorological disasters in both coastal provinces of Krabi and Nakhon Si Thammarat, vulnerable groups bemoaned the lack of money to construct decent and stronger houses. Often the need from the people is huge, but the local administration is limited in terms of sufficient data and budget. Residents attributed the increasing coastal erosion in the area to human error, mainly due to unending waves generated by boats since they are the main mode of transport. In other areas, however, residents indicated that they did not have serious effects from coastal erosion. Instead, the accretion of clay soil and too much sand in water bodies is threatening their livelihoods as it has made fishing bays too shallow.

Strict regulations by law and government affect fishermen a great deal. For example, people are required to register and get fishing permits. A majority of people in coastal areas are poor. When they go to the sea, they usually do not have any money, but they must pay tax. While it is good for fishermen to register, residents felt that the government needs to provide some assistance. In Krabi province, fishermen are not permitted to dock their boats in certain areas reserved for tourists. Trespassers are made to pay hefty penalty fees, and the fishermen are concerned as to why tourists are given preferential treatment at the expense of the local community. The participants in Krabi province indicated that the government needs to investigate and understand the impact of certain regulations on the community. The participants argued that some of the laws and regulations did not fit the context of certain communities, and before the announcement of any laws, only public hearings would ensure that people’s voices are heard.

Residents in Nakhon Si Thammarat indicated that corruption is rampant in their area, and it involves even the contractors. As such, most of the hard construction done in the coastal area is often substandard and will not last many years. In addition, construction done by different contractors is different in terms of materials used, not to mention the differences in the height of walls constructed.

The majority of people along the coastal areas do not have land licenses (title deeds) because the land or beach is under the responsibility of the Marine Office. As such, people must inform the officers should they want to construct something, and they must pay tax to the Marine Office. Therefore, there is no incentive to take the initiative and construct anything to protect their area from boat waves, because it attracts a penalty. Not every household has a boat, and during a disaster, those who own boats prioritize their families. Some people just chose not to be evacuated so that they could take care of their valuables. It was reported as unfortunate that some of those people who refused to be moved then posted on social media that the government did not care much. The majority of disabled persons do not like to be moved from their houses.

A major challenge highlighted by representatives from central, regional, and local government was that government agencies are responsible only for their specific areas. When budget is availed from the Ministry of Finance, different regulations are used in implementing the needed activities. For example, for disaster protection and mitigation, the regulations from Disaster Prevention and Mitigation Department take precedence. However, for compensating people following a disaster, regulations from the Ministry of Social Development and Human Security are followed.

The study is not without limitations, as qualitative strategy solely considers perspectives; as a result, it does not give a statistical representation. Moreover, it is challenging to demonstrate data rigidity since individual viewpoints frequently served as the basis of data for the study.

The recommendations are as follows: (1) Information sharing is important for effective collaboration among organizations during disaster response. In some cases, the Disaster Prevention and Mitigation Office may have some key information needed by the Mangrove Conservation Office about a specific issue. The two must share their information and work together. (2) Post-disaster compensation and benefits should be distributed and shared equitably among all affected groups. Suffering among member belonging to the same community is often the same. (3) The government should set aside a budget of about 100 million THB (USD 2.7 million) earmarked for studies in the coastal areas to examine the factors associated with coastal erosion, especially during the monsoon season. The participants felt that commissioning such a study would yield better results than paying 10 million THB (USD 0.27 million) for house construction because the houses are not permanent. Coastal erosion may change its pattern and direction at any given time. (4) Releasing information before disaster strikes is very important. The government, CSOs, and private sector players should leverage all available technologies to disseminate key information as this helps in saving lives, livelihoods, and money. (5) Plantation projects are key in mitigating the effects of coastal erosion. However, a plan for plantation projects for each area must be context-specific, considering the lifestyles of people in the area. In the future, any government agency may use such plans, without the need to analyze the situation again. (6) Every decision made by the government should consider the effects on the local people, and there is need for strong laws and regulations to preserve natural resources. (7) The Royal Forestry Department should desist from planting a certain type of mangrove forest (*Rhizophora mucronate*, or red mangrove) because it produces huge amounts of wastewater. (8) Each government agency should have a roadmap covering 5–20 years. The roadmap should lay out details of the agency’s initiatives and projects in a particular area. This would help to better forecast resources and identify opportunities for integrating projects to achieve efficiency.

## 5. Conclusions

In conclusion, each natural occurrence involves distinct encounters, feelings, and actions in coping with the hazards relating to flooding and erosion in coastal regions. The active involvement of governments and relevant actors together with the community in mapping, communication, and mobilization of resources about the event at all levels must include this aspect and be specifically adapted to it in a timely manner. These factors provide new angles for potential research directions in coping strategies for coastal disasters.

## Figures and Tables

**Figure 1 ijerph-19-12326-f001:**
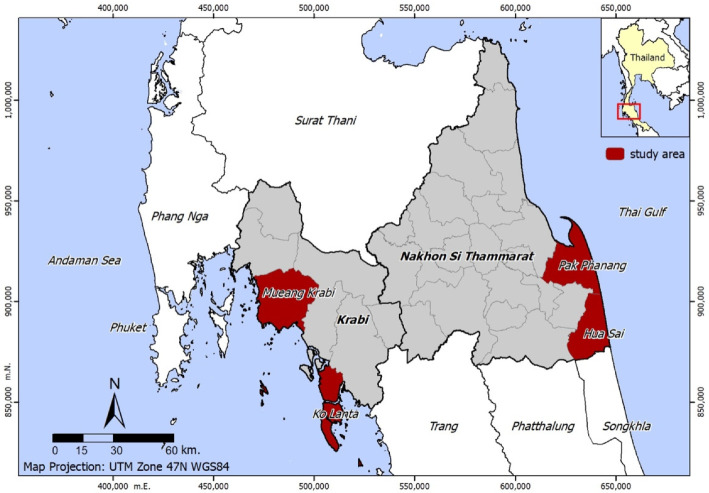
Location of the study area on the map of Krabi and Nakhon Si Thammarat provinces, Thailand.

**Figure 2 ijerph-19-12326-f002:**
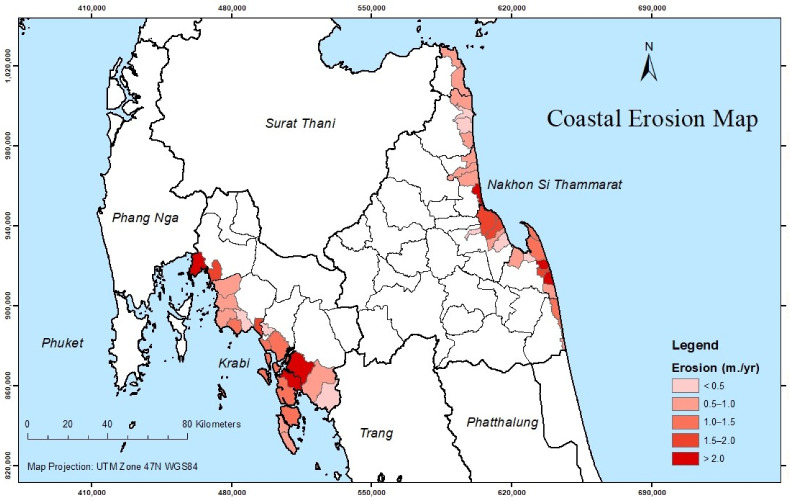
Coastal erosion rate along the Krabi and Nakhon Si Thammarat coasts, Thailand.

**Figure 3 ijerph-19-12326-f003:**
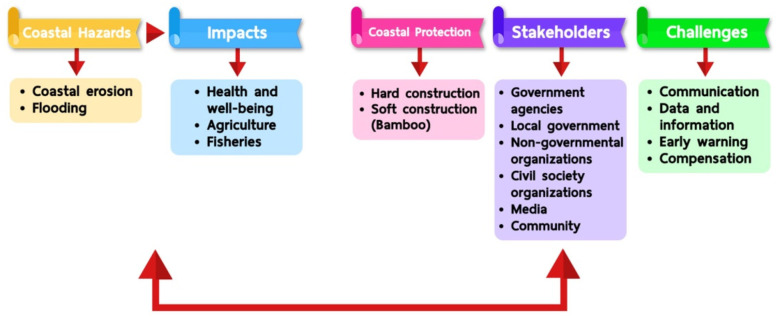
Local coping mechanisms and challenges due to climate change in Krabi and Nakhon Si Thammarat provinces, Thailand.

**Table 1 ijerph-19-12326-t001:** Total participants by province.

Participant Type	Krabi	Nakhon Si Thammarat	Total
Central, regional, and local government	6	6	12
Local fishery networks and civil society organizations (CSOs)	6	6	12
Non-governmental organizations (NGOs) and media organization	6	6	12
Community	6	6	12
People affected and vulnerable group	6	6	12
Total	30	30	60

## Data Availability

Not applicable.

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
