# Peer review of "Coastal Erosion and Flood Coping Mechanisms in Southern Thailand: A Qualitative Study"

_ijerph, 2022, doi:10.3390/ijerph191912326_

Round 1
Reviewer 1 Report
Dear authors, my corrections as follows:
1. Line 55: Here, enhance this argument adding the following references:
Morphodynamics of deltaic wetlands and implications for coastal ecosystems–A case study of Save River Delta, Mozambique.
Multidecadal biogeomorphic dynamics of a deltaic mangrove forest in Costa Rica
Bottom sediments affect Sonneratia mangrove forests in the prograding Mekong delta, Vietnam
2. Line 65: Check and add the following papers to improve this nice idea:
Flood risk index development at the municipal level in Costa Rica: A methodological framework
Future climate risk from compound events
Flood Risk-Related Research Trends in Latin America and the Caribbean
3. Figure 1: add more information in the caption, and add geographic coordinates
4. Line 109: have this approach have been applied before? how do you justify its use without any statistical method to support your results??? please add some references or examples of success of the method applied
5. Line 117: The paper is missing of graphs, maps, and tables showing a summary of your results in a more colorful and didactive way
6. Line 398: Please try to first relate, contrast and discuss your results with Thailand, after the tropics and developing countries, and finally with the whole world, trying to insert examples from all the continents. You are not posing your manuscript in an international framework.
Plase separate yor Discussion sections as the ones you used in Results and are coming from Methods
7. My detailed corrections in the attached PDF.
All the best

Author Response
Dear Reviewer 1,
Thank you for giving us the opportunity to submit a revised manuscript titled “Coastal Erosion and Flood Coping Mechanisms in Southern Thailand: A Qualitative Study” to International Journal of Environmental Research and Public Health. We appreciate the time and effort that you have dedicated to providing your valuable feedback on our manuscript. We are grateful to the reviewer for your insightful comments on our manuscript. We have been able to incorporate changes to reflect most of the suggestions provided by the reviewer. We have highlighted the changes within the revised manuscript.
Here is a point-by-point response to the reviewer’ comments and concerns.
Comments from Reviewer 1
- Comment 1: Line 55: Here, enhance this argument adding the following references:
- Morphodynamics of deltaic wetlands and implications for coastal ecosystems–A case study of Save River Delta, Mozambique.
- Multidecadal biogeomorphic dynamics of a deltaic mangrove forest in Costa Rica.
- Bottom sediments affect Sonneratia mangrove forests in the prograding Mekong delta, Vietnam.
Response: Thank you for your advice. We have added them in the revised manuscript – page 2, line 59 and references section.
- Comment 2: Line 65: Check and add the following papers to improve this nice idea:
- Flood risk index development at the municipal level in Costa Rica: A methodological framework.
- Future climate risk from compound events.
- Flood Risk-Related Research Trends in Latin America and the Caribbean.
Response: Thank you for your advice. We have added them in the revised manuscript – page 2, line 70 and references section.
- Comment 3: Figure 1: add more information in the caption and add geographic coordinates.
Response: Thank you for your advice. We have modified it in the revised manuscript – page 4, line 129-131.
- Comment 4: Line 109: have this approach have been applied before? how do you justify its use without any statistical method to support your results??? please add some references or examples of success of the method applied.
Response: This is a good question and thank you for bringing it up. We have modified it in the revised manuscript – page 5-6, line 148-154 and references section.
- Comment 5: Line 117: The paper is missing of graphs, maps, and tables showing a summary of your results in a more colorful and didactive way.
Response: Thank you for pointing this out. We agree with this comment. Therefore, we have modified the results in the revised manuscript – page 6, line 161-168.
- Comment 6: Line 398: Please try to first relate, contrast, and discuss your results with Thailand, after the tropics and developing countries, and finally with the whole world, trying to insert examples from all the continents. You are not posing your manuscript in an international framework.
Please separate your Discussion sections as the ones you used in Results and are coming from Methods.
Response: Thank you for your advice. We have modified the discussion in the revised manuscript – page 14-15, line 552-600.
- Comment 7: My detailed corrections in the attached PDF. Line 535: only 17 references are very limited for a research article.
Response: Thank you for your advice. We have added the references in the revised manuscript – page 17-18, references section.

Reviewer 2 Report
Review of “Coastal Erosion and Flood Coping Mechanisms in Southern Thailand: A Qualitative Study” by Langkulsen et al., submitted to International Journal of Environmental Research and Public Health, Section: Climate Change, Special Issue: Climate Driven Health Impacts
The manuscript examines the impact of coastal erosion and flooding on communities located in two area of southern Thailand. The study examines the response of local governments, CSOs, NGOs, communities and individuals to 13 different parameters to gauge what is important to them, what is lacking and what can be improved. While this is an important topic, there are flaws in this study that must be addressed before it can be published. Due to the amount of work that still needs to be done, I recommend rejection but encourage the authors to invest the time and resubmit.
First and foremost, while the English is not bad it does need to be improved. There are numerous grammatical errors throughout the manuscript.
Although this is a qualitative study, it still needs number and facts. The manuscript must include an extensive background not only on what coastal disasters exist in the area – storms, tsunamis, flooding – but also their frequency and intensity. In addition, since the topic is coastal erosion and flooding, a detailed introduction that includes areas the vulnerable to erosion and flooding (i.e. maps) as well as numbers – how many floods in past 50 years, their reoccurrence rates, how much land is being lost to erosion annually, etc. – is vital.
More specifically (and in addition to comments in the annotated manuscript):
The beginning of the Introduction is misleading and erroneous. The first two sentences are the setup and it would seem that the article’s main focus is extreme tides. In addition, it is erroneous - erosion and flooding are amplified by strong tidal activities but occur regardless of it.
Line 48 - Disappear or migrate landwards? The normal morphological process is that as sea levels rise, these features are pushed back landwards so that their width remains more-or-less the same. This is the foundation for the Bruun model. This becomes a problem however, when anthropogenic activities prevent this from happening.
Lines 48-49 – Aside from being important habitats, such features also act to protect the backshore from the effect of waves and storms.
Line 52 – Sea level has been rising since the end of the Last Glacial Maximum (LGM), so for at least 15000 years, at different rates without human intervention. Thus, sea level rise in itself does not only cause coastal erosion, but can also cause accretion (since there is sand on the beach) and the formation of new coastlines further inland. Again – this becomes a problem when human intervention prevents the formation of new beaches (by constructing roads, hotels, and infrastructure). It is then that humans need to react.
Study area
Since the article deals with coastal resilience, you need to provide a general geological/morphological/sedimentological background to the study area. Sandy or rocky beaches? Longshore transport? Erosion? Accretion? What has been done in terms of protection and intervention…?
Also – more quantitative descriptions are needed – how long is the coastline? What is the population living in the coastal areas? Why were these areas in Thailand chosen for this study and not others?
Methodology
How was the study conducted? Through questionnaires handed out to participants? Through interviews? In any case, you need to provide the questions that you asked.
Data analysis
How were the 13 items chosen? Some do not make sense – e.g. #2 - control measures to curb illegal fishing and coastal erosion – measure for curbing illegal fishing are different to those to curb coastal erosion. Why are these grouped together? Section 3.2 speaks about two separate issues – illegal fishing and coastal erosion – with no apparent connection between them. The connection exists (sort of – only through the allocation of funds and use of materials) but the way that it is written now does not show it at all. Also numerous studies are mentioned but no citations are provided for them.
Not enough information is provided – e.g. line 217: “Following the floods” - what floods? Do not assume the reader knows what you are talking about. What year? How much land was affected? How many people? This is a problem throughout the text (e.g. “the storm”). Again – the manuscript would benefit from maps showing the extent of flooding and areas affected by the storms mentioned in the text.
Results need to be included explicitly in the Discussion to see how you reached your conclusions. Statements such as “Also as uncovered through this study” line 461 need to be more explicit – Our result where………..(what you found)……..showed that….meaning….. The reader should not have to guess which of your results you are talking about nor have to try to figure out how you connect your results to the conclusion or point you are trying to make.
A lot of work has been done elsewhere in the world. A good example is Puerto Rico where community involvement is seen as a crucial element in disaster preparedness and recovery. Your manuscript could use with comparisons where successful programs have been implemented.
Figures – there is only one figure in the paper. This is a shame because it would benefit from illustrating some of the points made – show the construction in the Krabi province (45 degree blocks), the risk map of the Office of Disaster Preparedness and Mitigation. Line 402-403: map of areas prone to coastal erosion and their population would be helpful…
In summary, there is the potential for an interesting and important article but not in its current state. I also do not see the connection to the journal or to the theme of the special issue – both of which deal with public health. You can make the claim for mental wellbeing, but this should be stressed in the article itself.

Author Response
Dear Reviewer 2,
Thank you for giving us the opportunity to submit a revised manuscript titled “Coastal Erosion and Flood Coping Mechanisms in Southern Thailand: A Qualitative Study” to International Journal of Environmental Research and Public Health. We appreciate the time and effort that you have dedicated to providing your valuable feedback on our manuscript. We are grateful to the reviewer for your insightful comments on our manuscript. We have been able to incorporate changes to reflect most of the suggestions provided by the reviewer. We have highlighted the changes within the revised manuscript.
Here is a point-by-point response to the reviewer’ comments and concerns.
Comments from Reviewer 2
Review of “Coastal Erosion and Flood Coping Mechanisms in Southern Thailand: A Qualitative Study” by Langkulsen et al., submitted to International Journal of Environmental Research and Public Health, Section: Climate Change, Special Issue: Climate Driven Health Impacts
The manuscript examines the impact of coastal erosion and flooding on communities located in two area of southern Thailand. The study examines the response of local governments, CSOs, NGOs, communities and individuals to 13 different parameters to gauge what is important to them, what is lacking and what can be improved. While this is an important topic, there are flaws in this study that must be addressed before it can be published. Due to the amount of work that still needs to be done, I recommend rejection but encourage the authors to invest the time and resubmit.
- First and foremost, while the English is not bad it does need to be improved. There are numerous grammatical errors throughout the manuscript.
Response: Thank you for pointing this out. Proof reading and editing have been done to improve the English language.
- Although this is a qualitative study, it still needs number and facts. The manuscript must include an extensive background not only on what coastal disasters exist in the area – storms, tsunamis, flooding – but also their frequency and intensity. In addition, since the topic is coastal erosion and flooding, a detailed introduction that includes areas the vulnerable to erosion and flooding (i.e. maps) as well as numbers – how many floods in past 50 years, their reoccurrence rates, how much land is being lost to erosion annually, etc. – is vital.
Response: Agree. We have modified the introduction in the revised manuscript – page 3, line 100-102.
- More specifically (and in addition to comments in the annotated manuscript):
- The beginning of the Introduction is misleading and erroneous. The first two sentences are the setup and it would seem that the article’s main focus is extreme tides. In addition, it is erroneous - erosion and flooding are amplified by strong tidal activities but occur regardless of it.
Response: Thank you for your advice. We have modified the introduction in the revised manuscript – page 2, line 38-40.
- Line 48 - Disappear or migrate landwards? The normal morphological process is that as sea levels rise, these features are pushed back landwards so that their width remains more-or-less the same. This is the foundation for the Bruun model. This becomes a problem however, when anthropogenic activities prevent this from happening.
Response: Thank you for your advice. We have modified the introduction in the revised manuscript – page 2, line 47-50.
- Lines 48-49 – Aside from being important habitats, such features also act to protect the backshore from the effect of waves and storms.
Response: Thank you for your advice. We have modified the introduction in the revised manuscript – page 2, line 47-50.
- Line 52 – Sea level has been rising since the end of the Last Glacial Maximum (LGM), so for at least 15000 years, at different rates without human intervention. Thus, sea level rise in itself does not only cause coastal erosion, but can also cause accretion (since there is sand on the beach) and the formation of new coastlines further inland. Again – this becomes a problem when human intervention prevents the formation of new beaches (by constructing roads, hotels, and infrastructure). It is then that humans need to react.
Response: Agree. We have modified the introduction in the revised manuscript – page 2, line 53.
- Study area
- Since the article deals with coastal resilience, you need to provide a general geological/morphological/sedimentological background to the study area. Sandy or rocky beaches? Longshore transport? Erosion? Accretion? What has been done in terms of protection and intervention…?
Response: Thank you for pointing this out. We have modified the study area in the revised manuscript – page 3-4, line 113-124 and page 5, line 133-134.
- Also – more quantitative descriptions are needed – how long is the coastline? What is the population living in the coastal areas? Why were these areas in Thailand chosen for this study and not others?
Response: Thank you for pointing this out. We have modified the study area in the revised manuscript – page 3-4, line 113-124 and page 5, line 133-134.
- Methodology
- How was the study conducted? Through questionnaires handed out to participants? Through interviews? In any case, you need to provide the questions that you asked.
Response: Thank you for pointing this out. We have modified the methodology in the revised manuscript – page 5, line 138-145.
- Data analysis
- How were the 13 items chosen? Some do not make sense – e.g. #2 - control measures to curb illegal fishing and coastal erosion – measure for curbing illegal fishing are different to those to curb coastal erosion. Why are these grouped together? Section 3.2 speaks about two separate issues – illegal fishing and coastal erosion – with no apparent connection between them. The connection exists (sort of – only through the allocation of funds and use of materials) but the way that it is written now does not show it at all. Also numerous studies are mentioned but no citations are provided for them.
Response: Thank you for pointing this out. We have modified the data analysis in the revised manuscript – page 5-6, line 148-154.
- Not enough information is provided – e.g. line 217: “Following the floods” - what floods? Do not assume the reader knows what you are talking about. What year? How much land was affected? How many people? This is a problem throughout the text (e.g. “the storm”). Again – the manuscript would benefit from maps showing the extent of flooding and areas affected by the storms mentioned in the text.
Response: Thank you for pointing this out. We have modified the results in the revised manuscript – page 8, line 272-273.
- Results need to be included explicitly in the Discussion to see how you reached your conclusions. Statements such as “Also as uncovered through this study” line 461 need to be more explicit – Our result where………..(what you found)……..showed that….meaning….. The reader should not have to guess which of your results you are talking about nor have to try to figure out how you connect your results to the conclusion or point you are trying to make.
Response: Thank you for pointing this out. We have modified the discussion in the revised manuscript – page 14, line 581-589.
- A lot of work has been done elsewhere in the world. A good example is Puerto Rico where community involvement is seen as a crucial element in disaster preparedness and recovery. Your manuscript could use with comparisons where successful programs have been implemented.
Response: Thank you for pointing this out. We have modified the discussion in the revised manuscript – page 14, line 589-593.
- Figures – there is only one figure in the paper. This is a shame because it would benefit from illustrating some of the points made – show the construction in the Krabi province (45 degree blocks), the risk map of the Office of Disaster Preparedness and Mitigation. Line 402-403: map of areas prone to coastal erosion and their population would be helpful…
Response: Thank you for pointing this out. We have modified the methodology in the revised manuscript – page 5, line 133-134.
- In summary, there is the potential for an interesting and important article but not in its current state. I also do not see the connection to the journal or to the theme of the special issue – both of which deal with public health. You can make the claim for mental wellbeing, but this should be stressed in the article itself.
Response: Thank you for pointing this out. We have modified the introduction in the revised manuscript – page 3, line 94-99.

Round 2
Reviewer 1 Report
Dear authors, thank you to attend my corrections.
All the best.
Author Response
Dear Reviewer 1,
Thank you for giving us the opportunity to submit a revised manuscript titled “Coastal Erosion and Flood Coping Mechanisms in Southern Thailand: A Qualitative Study” to International Journal of Environmental Research and Public Health.
Reviewer 2 Report
I congratulate the authors for revising the manuscript in a quick and timely manner, while addressing the comments and concerns that I previously had. The manuscript has been improved drastically.
Please go over it carefully once again. The English has been improved but there are still numerous grammatical mistakes. I have tried to point then out and suggest corrections, but I have missed many, especially towards the end. Please see attached annotated review.

Author Response
Dear Reviewer 2,
Thank you for giving us the opportunity to submit a revised manuscript titled “Coastal Erosion and Flood Coping Mechanisms in Southern Thailand: A Qualitative Study” to International Journal of Environmental Research and Public Health. We appreciate the time and effort that you have dedicated to providing your valuable feedback on our manuscript. We are grateful to the reviewer for your insightful comments on our manuscript. We have been able to incorporate changes to reflect most of the suggestions provided by the reviewer. We have highlighted the changes within the revised manuscript.
Here is a point-by-point response to the reviewer’ comments and concerns.
Comments from Reviewer 2
Please go over it carefully once again. The English has been improved but there are still numerous grammatical mistakes. I have tried to point then out and suggest corrections, but I have missed many, especially towards the end. Please see attached annotated review.
Response: We agree with this and have incorporated your suggestion throughout the manuscript.
